# Population Pharmacokinetics and Significant Under-Dosing of Anti-Tuberculosis Medications in People with HIV and Critical Illness

**DOI:** 10.3390/antibiotics10060739

**Published:** 2021-06-18

**Authors:** Prakruti S. Rao, Christopher C. Moore, Amir A. Mbonde, Edwin Nuwagira, Patrick Orikiriza, Dan Nyehangane, Mohammad H. Al-Shaer, Charles A. Peloquin, Jean Gratz, Suporn Pholwat, Rinah Arinaitwe, Yap Boum, Juliet Mwanga-Amumpaire, Eric R. Houpt, Leonid Kagan, Scott K. Heysell, Conrad Muzoora

**Affiliations:** 1Division of Infectious Diseases and International Health, University of Virginia, Charlottesville, VA 22908, USA; pr6zu@virginia.edu (P.S.R.); ccm5u@virginia.edu (C.C.M.); jcg9p@virginia.edu (J.G.); spv4s@virginia.edu (S.P.); erh6k@virginia.edu (E.R.H.); 2Faculty of Medicine, Mbarara University of Science and Technology, Mbarara 1410, Uganda; aamir@must.ac.ug (A.A.M.); edwinuwagira@gmail.com (E.N.); rinah.arinaitwe@epicentre.msf.org (R.A.); Juliet.mwanga@epicentre.msf.org (J.M.-A.); conradmuzoora@must.ac.ug (C.M.); 3Department of Microbiology, University of Global Health Equity, Kigali 6955, Rwanda; porikiriza@ughe.org; 4Epicentre Mbarara Research Center, Mbarara 1956, Uganda; Dan.NYEHANGANE@epicentre.msf.org (D.N.); Yap.BOUM@epicentre.msf.org (Y.B.); 5Department of Pharmacotherapy and Translational Research, College of Pharmacy, University of Florida, Gainesville, FL 32610, USA; Mshaer@ufl.edu (M.H.A.-S.); peloquin@cop.ufl.edu (C.A.P.); 6Department of Pharmaceutics and Center of Excellence for Pharmaceutical Translational Research and Education, Ernest Mario School of Pharmacy, Rutgers, The State University of New Jersey, Piscataway, NJ 08854, USA; lkagan@pharmacy.rutgers.edu

**Keywords:** TB bacteremia, sepsis, meningitis, HIV, pharmacokinetics, population modeling, simulation

## Abstract

Critical illness from tuberculosis (TB) bloodstream infection results in a high case fatality rate for people living with human immunodeficiency virus (HIV). Critical illness can lead to altered pharmacokinetics and suboptimal drug exposures. We enrolled adults living with HIV and hospitalized with sepsis, with and without meningitis, in Mbarara, Uganda that were starting first-line anti-TB therapy. Serum was collected two weeks after enrollment at 1-, 2-, 4-, and 6-h post-dose and drug concentrations quantified by validated LC-MS/MS methods. Non-compartmental analyses were used to determine total drug exposure, and population pharmacokinetic modeling and simulations were performed to determine optimal dosages. Eighty-one participants were enrolled. Forty-nine completed pharmacokinetic testing: 18 (22%) died prior to testing, 13 (16%) were lost to follow-up and one had incomplete testing. Isoniazid had the lowest serum attainment, with only 4.1% achieving a target exposure over 24 h (AUC_0–24_) of 52 mg·h/L despite appropriate weight-based dosing. Simulations to reach target AUC_0–24_ found necessary doses of rifampin of 1800 mg, pyrazinamide of 2500–3000 mg, and for isoniazid 900 mg or higher. Given the high case fatality ratio of TB-related critical illness in this population, an early higher dose anti-TB therapy should be trialed.

## 1. Introduction

Critical illness, including sepsis from tuberculosis (TB) bloodstream infection or meningitis, is common among people living with HIV (PLWH) in TB prevalent settings [1,2]. Differing from populations elsewhere, in sub-Saharan Africa, *Mycobacterium tuberculosis* has been identified as the cause of 25–30% of bloodstream infections and is the leading cause of sepsis [3,4,5,6]. PLWH who develop sepsis from severe TB diseases are at a higher risk of dying compared to those without HIV [7]. Pharmacokinetics and pharmacodynamics (PK/PD) are important drivers of outcomes with TB treatment [8,9,10,11]. Among patients with pulmonary TB from South Africa, serum drug concentrations predicted key outcomes of microbiologic failure, death, or relapse [12]. In a separate study of PLWH and being treated for pulmonary TB in Uganda, the time to culture conversion (microbiological cure) was longer in patients with drug concentrations of isoniazid and rifampin that were below the expected peak concentrations [13]. These studies and others using various in vitro model systems have established target ranges for PK exposures of peak (C_max_) and total area under the concentration time curve (AUC_0–24_) for many anti-TB drugs [14,15]. Furthermore, increasing doses of first-line anti-TB drugs predictably increases serum exposures, and higher exposures can be targeted to improve microbial kill and cure [16,17].

Patients presenting with critical illness from other causes of sepsis often have decreased drug absorption, expanded drug volume of distribution, and altered drug–protein binding and/or oxidative metabolism that can result in significant PK variability [18]. Complicating our understanding of PK in TB-related critical illness is the considerable proportion that are treated empirically for TB after failing to respond to conventional antibacterials. PK exposures may differ among those with confirmed *M. tuberculosis-* related critical illness compared to those infected with more acutely pathogenic organisms, or when the causative pathogen is unknown. 

Thus, in this prospective cohort, we enrolled PLWH presenting with sepsis that were initiating anti-TB treatment to determine steady state PK exposures of rifampin, isoniazid, pyrazinamide, and ethambutol. We created a population pharmacokinetic model with simulations to assess different dosing approaches to attain minimum serum targets for these first-line drugs and to determine if any differences in exposures existed among people confirmed with *M. tuberculosis* compared to those with a clinical diagnosis of TB only. 

## 2. Results

A total of 81 patients were enrolled. There were 29 (36.3%) women and the median (interquartile range) age, weight, and MUAC (mid-upper arm circumference) were 36 (30–43.5) years, 52.5 (45.3–58) kgs, and 22 (20.4–24) cm, respectively (Table 1). Twenty-seven patients (33%) had concurrent meningitis. Seven (9%) had a prior history of TB. The median CD4 count was 169 cells/ml^3^ and 62 (76%) were on antiretroviral therapy (ART) prior to admission.

From all the diagnostic tests, there were 36 patients with confirmed TB (6 by blood TAC testing) and 45 with clinical TB. The initial PK visit at two weeks after enrollment was performed only in 50 (61.7%) patients as 18 (22.2%) had died, 5 (6.2%) withdrew, and 8 (9.9%) were lost to follow-up. One other patient had incomplete serum collection, bringing the total number of patients included in the final PK analyses to 49 (26 with confirmed TB, 23 with clinical TB) (Table 2).

### 2.1. Rifampin

A total of 172 non-zero concentration points from 49 patients were included for population modeling, with median (IQR) dosage, C_max_, and AUC_0–24_ of 10.1 (9–10.7) mg/kg, 3.8 (2.3–5.3) mg/L, and 21.7 (13.4–31.2) mg·h/L, respectively. Four (8.2%) patients reached serum C_max_ target ≥ 8 mg/L and eight (16.3%) reached AUC_0–24_ target ≥35.4 mg h/L. There was moderate variability in the observed data.

A one-compartment distribution model with a lag-time for oral absorption and an additive residual error was selected as a final model and provided a good description of the observed data. Residual variability was included on all parameters. Upon testing for covariates, weight was a significant covariate for clearance. The PK estimates along with interindividual variability for the final model are shown in Table 3. Goodness-of-fit plots and a visual predictive check (VPC) plot are shown in Figure 1. Individual patient simulations, done at 600, 900, 1200, 1500, and 1800 mg dose levels, are represented in Figure 2. The percentage of simulated subjects attaining the target C_max_ of 8 mg/L and the target AUC_0–24_ of 35.4 mg·h/L are described in Table 4.

### 2.2. Isoniazid

The population was described by a two-compartment model with a lag time for oral absorption, for a total of 173 non-zero concentration points. A proportional error model was used and the final model provided good description of observed data. Residual variability was included on all parameters. The median (IQR) dosage, C_max_, and AUC_0–24_ were 5 (4.5–5.4) mg/kg, 3.6 (2.3–4.6) mg/L and 22.5 (14.3–34) mg·h/L, respectively. Even with adequate weight-based dosing; only 4.10% reached the minimum AUC_0–24_ target of 52 mg·h/L and 63.3% reached C_max_ target ≥ 3 mg/L. NAT2 genotypic testing was available in 32 (65.3%) patients, with 29 being of the slow–intermediate type and the remaining three were rapid metabolizers. There were no significant differences in C_max_ or AUC_0–24_ values between the slow–intermediate type and the few rapid metabolizers, and thus the NAT2 genotype was not included among the covariates. Screening for age, sex, weight, MUAC, and TB diagnostic categorization yielded a significant covariate effect of MUAC on the volume of distribution of central compartment (V). The final two-compartment model had PK estimates as represented in Table 3. Goodness-of-fit plots and VPC are shown in Figure 3. Individual patient simulation was done at 300 mg, 450 mg, 600 mg, 750 mg, and 900 mg (Figure 4). The percentage of simulated subjects attaining the target C_max_ of 3 mg/L and the target AUC_0–24_ of 52 mg·h/L are described in Table 4.

### 2.3. Pyrazinamide

One-compartment distribution model with a lag-time for oral absorption and additive residual error was selected as a final model and provided a good description of the observed 173 concentration points. The residual variability was included on all parameters. The median (IQR) dosage, C_max_, and AUC_0–24_ were 25.4 (23–28) mg/kg, 34 (28.3–44) mg/L, and 351 (237.1–477.9) mg·h/L, respectively, with 87.8% reaching the C_max_ threshold ≥ 20 mg/L and 38.8% reaching AUC_0–24_ target ≥ 363 mg·h/L. None of the covariates were significant. PK parameter estimates of the final model are shown in Table 3. Goodness-of-fit plots and VPC are shown in Figure 5. Individual patient simulation was completed at 1000 mg, 1500 mg, 2000 mg, 2500 mg, and 3000 mg dose levels (Figure 6). The percentage of simulated subjects attaining the target C_max_ of 20 mg/L and the target AUC_0–24_ of 363 mg·h/L are described in Table 4.

### 2.4. Ethambutol

A total of 173 concentration points from 49 patients were included for population modeling, with a median (IQR) dosage of 18.4 (16.5–19.6) mg/L and 30.6% attained target C_max_ ≥ 2 mg/L with median 1.8 (1.3–2.2) mg/L. The median AUC_0–24_ was 14.3 (10.6–26.6). A one-compartment distribution model with a lag-time for oral absorption and additive residual error was selected as a final model and provided a good description of the observed data. None of the covariates screened were significant. PK estimates and interindividual variability for the final model are shown in Table 3. Goodness-of-fit plots and VPC based on 1000 Monte Carlo simulations are shown in Figure 7. Individual patient simulation (Figure 8) was done at 800 mg, 1200 mg, 1600 mg, 2000 mg, and 2400 mg. As no target value has been described for ethambutol AUC_0–24_, only the percentage of simulated patients reaching target C_max_ has been shown in Table 4. 

## 3. Discussion

PLWH and presenting with critical illness that were treated for TB had considerably altered PK with the majority failing to reach one or more PK targets for rifampin, isoniazid, pyrazinamide and ethambutol. To our knowledge, this is the first PK study of anti-TB drugs among PLWH with sepsis physiology from a TB endemic setting. Population PK modeling and individual patient simulation suggest necessary dosages to reach the target total serum exposure, AUC_0–24_, would require a tripling of the typical rifampin dose to 1800 mg, and a modest increase in pyrazinamide dosages of 2500 to 3000 mg. Surprisingly, simulating for a daily dose as high as 900 mg for isoniazid, only 4.5% reached the target AUC_0–24_. While the target AUC_0–24_ for ethambutol has not been established, dosing from 1600 and 2000 mg only modestly increased the proportion attaining a target peak concentration.

Importantly, nearly a quarter of all PLWH and those treated for TB-related critical illness died before reaching the two-week PK visit and presumed steady-state kinetics. An expected lower proportion of patients with confirmed TB died before the two-week PK visit compared to those without microbiological confirmation of TB. While the high and early case fatality rate is not dissimilar to other cohorts with HIV and critical illness in Uganda [19], we were unable to determine if there were PK differences among those with early mortality and those who survived to PK testing. Given that the pathophysiology of severe sepsis leads to altered plasma drug binding capabilities and decreased drug perfusion to tissues, and impaired metabolism and clearance due to liver and kidney injury [18], it is possible that those patients with early mortality may have had unexpectedly higher serum exposures than were measured in the cohort of survivors. Further studies of anti-TB PK in critical illness should assay drug concentrations within days of TB treatment initiation and again at a steady state.

Rifampin is a potent antituberculous drug that produces antimicrobial activity by inhibition of DNA-dependent RNA polymerase, and the absorption can be affected by the *SLCO1B1* gene [20]. Isoniazid is rapidly absorbed from the gastrointestinal tract and the serum concentrations can be affected by the *NAT2* phenotype classification [21]. Pyrazinamide is also readily absorbed, and all drugs act in a concentration-dependent manner; thus, higher serum exposures produce a greater antibacterial effect. However, the considerable PK variability observed in this cohort, and the high number failing to reach exposures for rifampin and isoniazid in particular, raise concern that even among those surviving critical illness, such low exposures may predispose to other longer term poor outcomes, such as acquired *M. tuberculosis* drug resistance, which have been more commonly observed with HIV co-infection [13,22]. It is therefore all the more important to determine if any measurable predictors at TB treatment initiation can predict PK exposures or improve upon a population PK model of anti-TB medications in PLWH and critical illness. In certain cohorts of ambulatory people with pulmonary TB, with or without HIV, drug dose, sex and low weight have been found to be important predictors of reduced rifampin and pyrazinamide exposure [23,24]. Isoniazid serum exposure has been better able to be explained by the NAT2 genotype and assignment of the phenotypic category (slow-intermediate and rapid metabolizers) in other cohorts due to its influence on isoniazid clearance [25], but not all patients in this study had NAT2 genotype testing, as this was a saliva test that required active patient participation, and only three participants were ultimately categorized as rapid metabolizers. We suspect the overall lack of covariate influence to be a result of too sparse a sampling strategy in a population with considerably altered physiology due to the sepsis state, and further models may be improved by more frequent sampling in the dosing interval. Furthermore, collection of biological markers for the extent of tissue damage such as lactate, which was only infrequently measured during routine care in this study, or immune profiles that may be more common in PLWH and disseminated TB disease, could be of further discriminatory benefit [26,27]. 

While this study advances our understanding about the extent of PK variability and under-dosing for rifampin and isoniazid in PLWH and critical illness, exposure targets are nonetheless extrapolated from in vitro models that have only been replicated in limited cohorts of ambulatory patients with pulmonary TB. Different exposure targets may be necessary for PLWH and critical illness from TB-related sepsis. Such targets may not be fully elucidated without controlled trials of immediate TB treatment among PLWH from TB-endemic settings that present with sepsis and who may not yet have a microbiologically confirmed diagnosis of TB, particularly if survival is the outcome of interest. To help design such studies, simulations were conducted for widely available dosages of each drug, with the highest median being 34 mg/kg of rifampin, 17 mg/kg of isoniazid, 57 mg/kg of pyrazinamide, and 45 mg/kg of ethambutol. Recent trials have demonstrated a tolerable use of higher doses of rifampin up to 40 mg/kg in pulmonary TB patients with mild to moderate related adverse effects [28,29,30]. Similarly, patients treated with higher isoniazid doses between 15 and 20 mg/kg have had improved outcomes with faster culture conversion in rifampin-resistant anti-TB regimens [31], yet the essentiality of isoniazid has been more difficult to demonstrate for rifampin-susceptible TB [32]. Lastly, while pyrazinamide would not appear to require a significant dose increase in this population of PLWH and critical illness in Uganda, a meta-analysis of 29 controlled studies suggested that doses up to 60 mg/kg of pyrazinamide did not lead to higher rates of hepatotoxicity [33]. 

In summary, death from critical illness among PLWH was common within the first two weeks of starting TB treatment, necessitating earlier interventions. Further studies of anti-TB PK in critical illness should measure the dynamics of drug concentrations from TB treatment initiation to steady-state and consider markers of tissue damage or severe immune perturbation as covariates. Regardless, PK modeling from this population of PLWH and critical illness suggest that higher doses of rifampin and considerably higher doses of isoniazid are required to reach conventional serum targets.

## 4. Materials and Methods

### 4.1. Patient Population

PLWH presenting with sepsis [34], defined as two or more systemic inflammatory response syndrome criteria and clinical suspicion of infection, with or without the additional clinical suspicion for TB meningitis as defined by the International TB Meningitis Workshop Consensus Case Definitions for TB Meningitis, and starting on first-line therapy for presumed drug-susceptible TB, were enrolled at Mbarara Regional Referral Hospital, Uganda. Males and non-pregnant females over the age of 18 were eligible to participate and informed consent was collected from patients who were alert, or from guardians in the case of patients presenting with altered levels of consciousness. A target of 80 consecutive patients meeting eligibility were enrolled in order for 50 people to undergo PK testing given expected mortality and loss to follow-up. The protocol was approved by institutional review boards of Mbarara University of Science and Technology, Uganda Council for Science and Technology, and the University of Virginia (ClinicalTrials.gov identifier NCT03559582, Charlottesville, VA, USA).

At study enrollment, TB diagnostics performed included sputum Xpert MTB/RIF, sputum mycobacterial cultures, mycobacterial blood cultures, and urine lipoarabinomannan (LAM). Most patients that could expectorate had more than one TB diagnostic performed. Whole blood was also collected for polymerase chain reaction (PCR)-based multiplexed TaqMan Array Card (TAC) to detect 43 common bacterial, fungal, viral, and parasitic targets, including *M. tuberculosis*. The TAC procedures are described in detail elsewhere [5,35]. TAC testing was performed in a batch and the results were not available for informing clinical care. Patients were categorized post hoc into two groups based upon availability of microbiology results: confirmed TB (TB identified by at least one diagnostic modality) or clinical TB (no diagnostic modality positive for TB). Antiretroviral therapy (ART) status and CD4 profiles, and other routine laboratory parameters were collected from medical charts if ordered by treating clinicians. Patients with suspicion of meningitis had cerebrospinal fluid tested to obtain a cell count profile prior to study entry. Prior to TB treatment initiation, ceftriaxone was the standard of care for empiric therapy of sepsis. 

### 4.2. NAT2 Testing

N-acetyltransferase-2 (NAT2) phenotype (intermediate/slow versus rapid metabolizer) was also determined by NAT2 genotyping for potential influence on isoniazid kinetics. For NAT2 targets, saliva samples were collected when participants were able to participate for PCR testing of the host targets with targets procured from Thermo Fisher (Thermo Fisher, Waltham, MA, USA) in a custom-designed plate with primers and probes pre-assembled in each well. NAT2 targets included: c.341 (C/T; rs1801280), c.590 (A/G; rs1799930), and c.803 (A/G; rs1208). Individual NAT2 genotypes at positions c.341 and c.590 were combined into haplotypes and then converted to NAT2-predicted acetylator phenotypes of “rapid” or “intermediate/slow” according to the indicated phenotypes available at the database of human NAT2 alleles hosted at the Democritus University of Thrace (Vasilissis Sofias, Greece) (https://nat.mbg.duth.gr/, accessed 18 December 2020).

### 4.3. Anti-TB Therapy and PK Study Methods

Participants were empirically administered first-line anti TB regimen that included rifampin, isoniazid, pyrazinamide, and ethambutol as weight-based fixed-dose combinations once daily. The first PK visit was conducted two weeks after the patients were started on anti-TB therapy to allow for steady-state kinetics. Sparse sampling strategies were used to collect venous blood at 1-, 2-, 4-, and 6-h post-dose. Medication was administered orally as directly observed therapy for the PK visit. Once blood was collected, the vials were centrifuged to separate serum, which was immediately frozen at −80 °C until batch shipment to University of Florida, USA (Gainesville, VA, USA), where concentrations of all four drugs were quantified using validated liquid chromatography-tandem mass spectrometry procedures, including prior testing for confirmation of milligram quantity in fixed dose preparations [36]. The methods were validated over the concentration ranges 0.4 to 20 µg/ml for isoniazid, 0.5 to 50 µg/ml for rifampin, 2 to 100 µg/ml for pyrazinamide, and 0.2 to 10 µg/ml for ethambutol [37].

### 4.4. Population Pharmacokinetics, Model Development, and Simulation

Individual PK profiles for four drugs were estimated using a noncompartmental approach in Phoenix WinNonlin version 8.0 (Certara, Princeton, NJ, USA) and C_max_ (maximal serum concentration) and AUC_0–24_ (areas under the serum concentration-time curve from time zero to 24 h, estimated in those with 6-h values within the elimination phase) were calculated and reported as median (interquartile range). 

Population PK models were developed (separately for each drug) using a nonlinear mixed-effects modeling approach (Phoenix NLME), and a first-order conditional estimation-extended least squares algorithm was used for model development. One- and two-compartment distribution models with linear elimination and first-order absorption (with and without lag-time) were tested. Interindividual variability was included using exponential function. Additive, proportional, and combined residual error models were tested. Model selection was based on a visual inspection of model fits and diagnostic plots, precision of parameter estimates, and Akaike and Schwartz information criteria. Available covariates included age, sex, total body weight, mid-upper arm circumference (MUAC), TB diagnostic group (confirmed and unconfirmed), and for isoniazid modeling, the NAT2 phenotype (intermediate/slow versus rapid metabolizer). A stepwise covariate search run mode was used for the evaluation of covariates (*p* = 0.05 forward/*p* = 0.01 backward). Covariates were tested using an exponential function in multiplicative fashion. To evaluate the robustness of the final model, visual predictive checks based on 1000 Monte Carlo simulations were performed.

Next, the final models were used to generate simulated pharmacokinetic profiles for individual patients (using individual PK parameters and covariates; and assuming dose-independent PK) at various available doses. The doses for rifampin were 600 mg, 900 mg, 1200 mg, 1500 mg and 1800 mg; for isoniazid at 300 mg, 450 mg, 600 mg, 750 mg and 900 mg; for pyrazinamide at 1000 mg, 1500 mg, 2000 mg, 2500 mg and 3000 mg; and for ethambutol at 800 mg, 1200 mg, 1600 mg, 2000 mg and 2400 mg. Individual simulated profiles were used to calculate C_max_ and AUC_0–24_. Box and whisker plots were used to illustrate simulated concentrations and the percentage of simulated patients attaining threshold exposure [15] for each drug. 

## Figures and Tables

**Figure 1 antibiotics-10-00739-f001:**
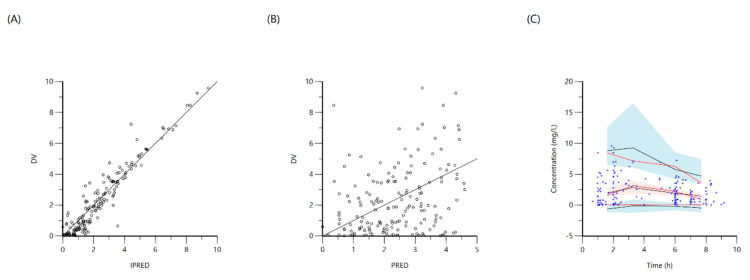
Goodness-of-fit plots for final population models of rifampin. Plot (**A**) represents individual predicted concentrations (IPRED) versus observed concentrations (DV) whereas, plot (**B**) represents population-predicted concentrations (PRED) versus observed concentrations (DV). Plot (**C**), Visual predictive check (VPC) for concentrations of rifampin versus time based on 1000 Monte Carlo simulations. Blue circles represent observed rifampin concentrations, solid red lines represent the 5th, 50th and 95th percentiles of the observed concentrations, solid black lines represent the 5th, 50th and 95th percentiles of the simulated concentrations, and the shaded area represents the 90% CI of the 5th, 50th and 95th percentiles of the simulated concentrations.

**Figure 2 antibiotics-10-00739-f002:**
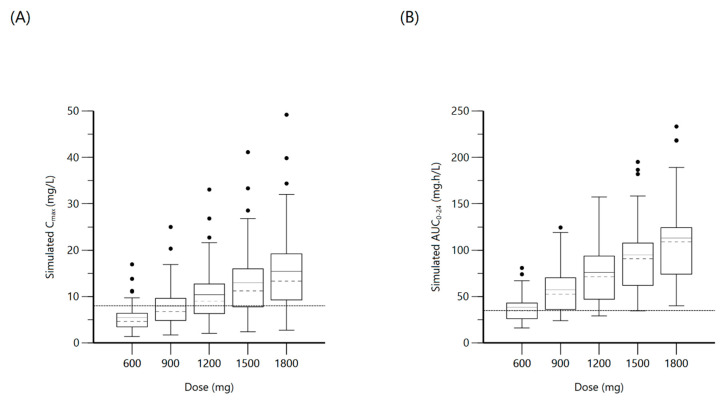
Plot (**A**) represents simulated rifampin C_max_ at various doses with the dashed line indicating the minimum threshold of 8 mg/L. Plot (**B**) represents simulated rifampin AUC_0–24_ at various doses with the dashed line indicating the minimum threshold of 35.4 mg·h/L.

**Figure 3 antibiotics-10-00739-f003:**
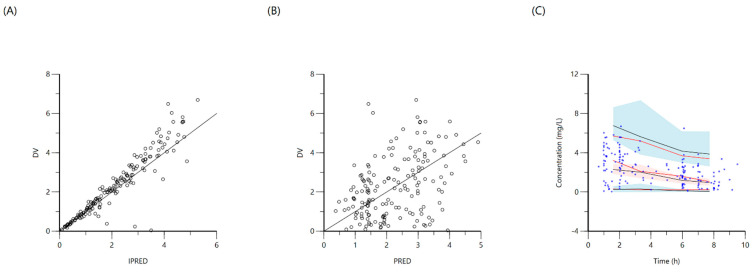
Goodness-of-fit plots for final population models of isoniazid. Plot (**A**) represents individual predicted concentrations (IPRED) versus observed concentrations (DV), whereas plot (**B**) represents population-predicted concentrations (PRED) versus observed concentrations (DV). Plot (**C**), visual predictive check (VPC) for concentrations of isoniazid versus time based on 1000 Monte Carlo simulations. The blue circles represent observed isoniazid concentrations; solid red lines represent the 5th, 50th and 95th percentiles of the observed concentrations; solid black lines represent the 5th, 50th and 95th percentiles of the simulated concentrations; and the shaded area represent the 90% CI of the 5th, 50th and 95th percentiles of the simulated concentrations.

**Figure 4 antibiotics-10-00739-f004:**
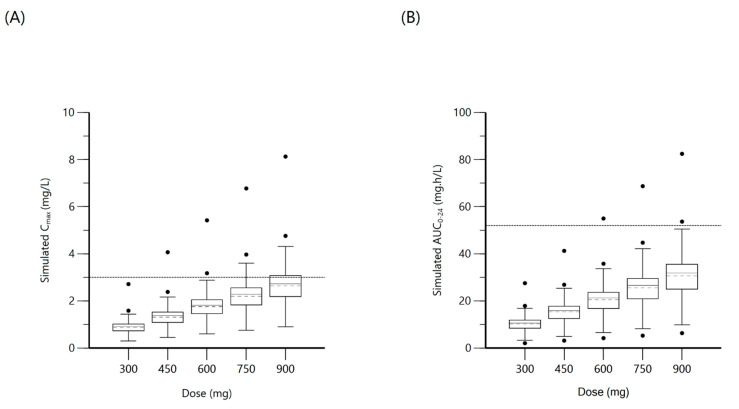
Plot (**A**) represents simulated isoniazid C_max_ at various doses with the dashed line indicating the minimum threshold of 3 mg/L. Plot (**B**) represents simulated isoniazid AUC_0–24_ at various doses with the dashed line indicating the minimum threshold of 52 mg·h/L.

**Figure 5 antibiotics-10-00739-f005:**
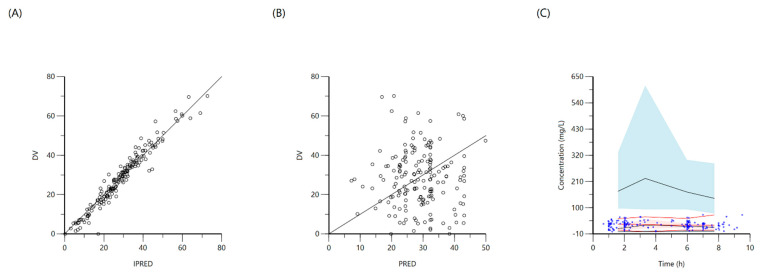
Goodness-of-fit plots for final population models of pyrazinamide. Plot (**A**) represents individual predicted concentrations (IPRED) versus observed concentrations (DV), whereas plot (**B**) represents population-predicted concentrations (PRED) versus observed concentrations (DV). Plot (**C**), visual predictive check (VPC) for concentrations of pyrazinamide versus time based on 1000 Monte Carlo simulations. Blue circles represent observed pyrazinamide concentrations, solid red lines represent the 5th, 50th and 95th percentiles of the observed concentrations, solid black lines represent the 5th, 50th and 95th percentiles of the simulated concentrations, and the shaded area represent the 90% CI of the 5th, 50th and 95th percentiles of the simulated concentrations.

**Figure 6 antibiotics-10-00739-f006:**
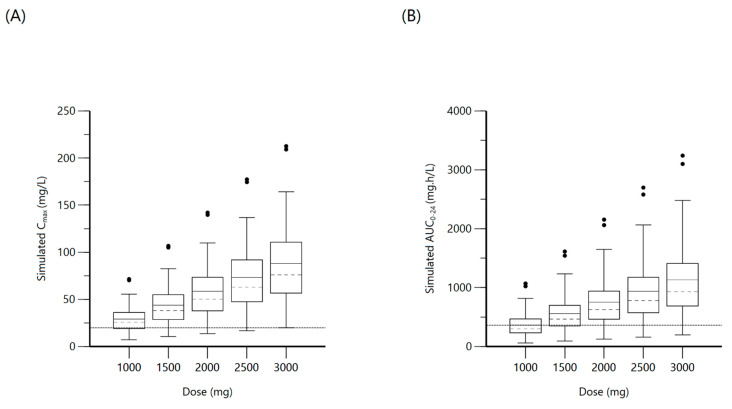
Plot (**A**) represents simulated pyrazinamide C_max_ at various doses with the dashed line indicating the minimum threshold of 20 mg/L. Plot (**B**) represents simulated pyrazinamide AUC_0–24_ at various doses with the dashed line indicating the minimum threshold of 363 mg·h/L.

**Figure 7 antibiotics-10-00739-f007:**
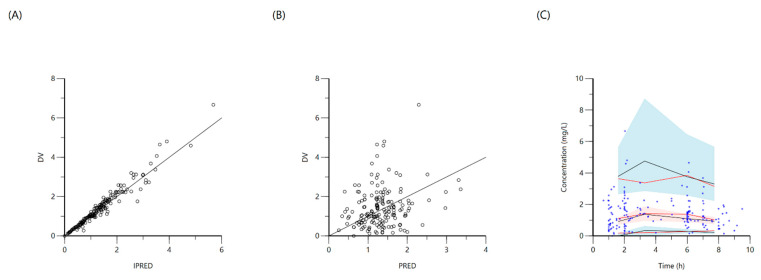
Goodness-of-fit plots for final population models of ethambutol. Plot (**A**) represents individual predicted concentrations (IPRED) versus observed concentrations (DV), whereas plot (**B**) represents population predicted concentrations (PRED) versus observed concentrations (DV). Plot (**C**), visual predictive check (VPC) for concentrations of ethambutol versus time based on 1000 Monte Carlo simulations. Blue circles represent observed ethambutol concentrations, solid red lines represent the 5th, 50th and 95th percentiles of the observed concentrations, solid black lines represent the 5th, 50th and 95th percentiles of the simulated concentrations, and the shaded areas represent the 90% CI of the 5th, 50th and 95th percentiles of the simulated concentrations.

**Figure 8 antibiotics-10-00739-f008:**
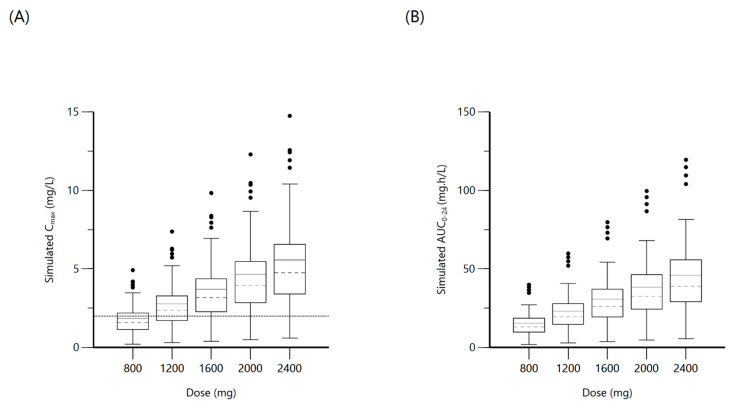
Plot (**A**) represents simulated ethambutol C_max_ at various doses with the dashed line indicating the minimum threshold of 2 mg/L. Plot (**B**) represents simulated ethambutol AUC_0–24_ at various doses.

**Table 1 antibiotics-10-00739-t001:** Demographic and clinical characteristics among people living with HIV, presenting with critical illness and treated for tuberculosis. MUAC: mid-upper-arm circumference. TB: Tuberculosis.

Characteristic	*N* (%) or Median (IQR) *N* = 81
Age (years)	36 (30–43.5)
Female	29 (36.3%)
Weight (kg)	52.5 (45.3–58)
MUAC (cm)	22 (20.4–24)
CD4 count (cells/mL^3^)	169
Prior TB	7 (9%)
Concurrent meningitis	27 (33%)
Duration of Illness (days)	30 (14–90)
Cough >2 weeks	63 (79%)
Fever >2 weeks	75 (94%)
Night sweats >2 weeks	67 (86%)
Loss of appetite >2 weeks	78 (96.3%)
Loss of wt >2 weeks	48 (58.3%)

**Table 2 antibiotics-10-00739-t002:** Vital status at week 2 pharmacokinetic testing. TB: Tuberculosis.

Vital Status at Week 2 PK Testing	All *N* = 81	Confirmed TB *N* = 36	Clinical TB *N* = 45
Survived (%N)	49 (60%)	26 (72%)	23 (51%)
Sepsis	32	21	11
Meningitis	17	5	12
Died (%N)	18 (23%)	5 (14%)	13 (29%)
Sepsis	13	4	9
Meningitis	5	1	4
Lost to follow-up, withdrew or incomplete PK testing (%N)	14 (17%)	5 (17%)	9 (20%)
Sepsis	9	5	4
Meningitis	5	0	5

**Table 3 antibiotics-10-00739-t003:** Pharmacokinetic estimates for the final models of rifampin, isoniazid, pyrazinamide, and ethambutol. Ka, absorption rate constant; V, volume of distribution in the central compartment; V2, volume of distribution of the peripheral compartment; Cl, systemic clearance; Cl2, intercompartmental clearance; Tlag, lag-time. %CV, coefficient of variation or precision; IIV, inter-individual variation.

Drug	Parameter	Final Model	
		Estimate (%CV)	IIV (Shrinkage)
Rifampin	Ka (1/h)	0.3 (13.5)	0.01 (0.9)
	V (L)	25.3 (33.3)	3.2 (0.2)
	Cl (L/h)	0.1 (49.6)	0.2 (0.3)
	Tlag (h)	0.9 (9.1)	0.003 (0.8)
Isoniazid	Ka (1/h)	0.9 (0.9)	0.9 (0.3)
	V (L)	2.9 (3)	0.07 (0.6)
	V2 (L)	32.5 (5.7)	0.5 (0.7)
	Cl (L/h)	9.2 (8.7)	0.8 (0.1)
	Cl2 (L/h)	9.6 (3)	0.0001 (0.9)
	Tlag (h)	0.4 (8.6)	0.5 (0.5)
Pyrazinamide	Ka (1/h)	0.08 (13)	0.1-0.11
	V (L)	1.5 (16.3)	0.9 (0.4)
	Cl (L/h)	2.6 (14.5)	0.4 (0.05)
	Tlag (h)	0.2 (7.1)	1.3 (0.4)
Ethambutol	Ka (1/h)	0.15 (21.8)	0.6 (0.4)
	V (L)	75.17 (23.7)	0.4 (0.1)
	Cl (L/h)	51.6 (12.2)	0.3 (0.3)
	Tlag (h)	0.4 (12.3)	0.6 (0.4)

**Table 4 antibiotics-10-00739-t004:** Summary of pharmacokinetic measurements and percentage attaining target at various simulated doses. C_max_ = peak serum concentration. AUC_0–24_ = serum total area under the concentration time curve. IQR= interquartile range.

Drug	Simulated Dose (mg)	Target	C_max_ (mg/L)		AUC_0–24_ (mg·h/L)	
			Median (IQR)	Attaining Target (%)	Median (IQR)	Attaining Target (%)
Rifampin	600	C_max_: 8 mg/L	4.6 (3.4–6.4)	16.30	34.5 (26.3–42.2)	43.6
	900		6.8 (4.8–9.6)	34.7	52.3 (36.6–63.5)	78
	1200		9 (6.3–12.7)	57.1	71.2 (48.1–84)	92.7
	1500	AUC_0–24_: 35.4 mg·h/L	11.2 (7.9–16)	73.5	90.6 (66.7–105.6)	97.7
	1800		13.3 (9.5–19.2)	81.6	108.8 (79–123.9)	100
Isoniazid	300	C_max_: 3 mg/L	0.9 (0.7–1)	0	10.2 (8.7–11.8)	0
	450		1.3 (1.1–1.5)	0	15.3 (13.1–17.7)	0
	600		1.7 (1.5–2)	4.1	20.5 (17.4–23.6)	2.3
	750	AUC_0–24_: 52 mg·h/L	2.2 (1.8–2.5)	8.2	25.6 (21.7–29-6)	2.3
	900		2.6 (2.2–3.1)	28.6	30.7 (26.1–35.5)	4.5
Pyrazinamide	1000	C_max_: 20 mg/L	25.7 (19.1–36)	69.4	303.8 (231.6–466.1)	42.9
	1500		38 (28.7–54.3)	89.8	466 (351.1–702.1)	71.4
	2000		50.4 (38.1–72.6)	91.8	624.7 (469.5–934.8)	85.1
	2500	AUC_0–24_: 363 mg·h/L	63.1 (47.4–90.8)	96	778.5 (588–1167.8)	89.6
	3000		75.8 (56.7–109.2)	98	933 (706.6–1403.9)	91.7
Ethambutol	800	C_max_: 2 mg/L	1.6 (1.2–2.2)	32.6	13 (10–18.6)	
	1200		2.4 (1.7–3.3)	63.2	19.5 (15–28)	
	1600		3.2 (2.3–4.4)	89.8	26 (20–37.1)	
	2000		3.9 (2.9–5.5)	93.8	32.5 (25–46.4)	
	2400		4.7 (3.5–6.5)	93.8	39 (30–55.7)	

## Data Availability

The data presented in this study are available on request from the corresponding author. The data are not publicly available due to original data use agreements but may be available in the future.

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
