# Peer review of "Population Pharmacokinetics and Significant Under-Dosing of Anti-Tuberculosis Medications in People with HIV and Critical Illness"

_antibiotics, 2021, doi:10.3390/antibiotics10060739_

Round 1

Reviewer 1 Report

The article deals with tuberculosis, a disease that has a long history, as well as possible medications. Treatment with 3 different drugs and their effectiveness are analyzed.
the safety methodology for the study is adequate, but some aspect can be improved.
In the paper there are too many abbreviations, it is convenient that the authors make a section of abbreviations.
In addition, the results of the dosage of each antibiotic are explained, but no explanation is given based on the pharmacological action of them. By introducing this, it would not only be an article where some observations are exposed, but a dosage could be thought about depending on the patients that belong to different region.

Author Response

Thank you for the comments.

Point 1: In the paper there are too many abbreviations, it is convenient that the authors make a section of abbreviations.

Response 1: An abbreviation section has been added after the abstract.

Point 2: In addition, the results of the dosage of each antibiotic are explained, but no explanation is given based on the pharmacological action of them. By introducing this, it would not only be an article where some observations are exposed, but a dosage could be thought about depending on the patients that belong to different region.

Response 2: A paragraph regarding the pharmacological effects of all four first-line drugs is included in the discussion (lines 252-257). The higher dosages indicated in the manuscript were a result of simulations conducted and these doses were not actually tested in our patient population.

Reviewer 2 Report

The research performed by the authors is significant, and can be a warning sign for people working in the treatment of TB with PLWH. The desciption of the results was convincing, however, the discussion could have gone more deeply into the analysis of the distinct groups e.g. sex, weight, age. Moreover, I could not find the desciption of the LCMSMS methods mentioned in the manuscript.

Author Response

Thank you for the comments.

Point 1: The desciption of the results was convincing, however, the discussion could have gone more deeply into the analysis of the distinct groups e.g. sex, weight, age.

Response 1: We conducted covariate analyses to determine the impact of sex, age, weight on serum concentrations of all four drugs. The effects of these parameters were not significant, and we hypothesized that it could be due to the small number of patients surviving up to the pharmacokinetic visit to undergo serum collection, and this reason has been mentioned in lines 272-275 in discussion section.   

Point 2: Moreover, I could not find the desciption of the LCMSMS methods mentioned in the manuscript.

Response 2: We have added a description of the methods along with a citation (lines 360-362).